
# Improvement in cloud retrievals from VIIRS through the use of infrared absorption channels constructed from VIIRS-CrIS data fusion

Yue Li[1], Bryan A. Baum[2], Andrew K. Heidinger[3], W. Paul Menzel[1], Elisabeth Weisz[1]

[1]Cooperative Institute for Meteorological Satellite Studies, University of Wisconsin-Madison, Madison, Wisconsin, USA
[2]Science and Technology Corporation, Madison, Wisconsin, USA
[3]NOAA/NESDIS/Center for Satellite Applications and Research, Madison, Wisconsin, USA

*Correspondence to*: Yue Li (yue.li@ssec.wisc.edu)

**Abstract**. Retrieval of semitransparent ice cloud properties from the Visible Infrared Imaging Radiometer Suite (VIIRS) satellite sensor on the Suomi-NPP and NOAA-20 platforms is challenging due to the absence of infrared (IR) water vapor and $CO_2$ absorption channels. However, on these platforms, there is a companion sensor called the Crosstrack Infrared Sounder (CrIS) that provides these spectral measurements, but at a lower spatial resolution (~15 km at nadir). To

mitigate the lack of VIIRS spectral measurements in these IR absorption channels, recent studies suggest an approach to supplement VIIRS measurements by fusion of the imager and sounder data. In particular, Weisz et al. (2017) demonstrate a method to construct IR water vapor and $CO_2$ absorption channel radiances for VIIRS at 750m spatial resolution. Based on these constructed channels for both Suomi-NPP and NOAA-20, this study evaluates three cloud properties – cloud

mask, cloud thermodynamic phase, and cloud top height – through comparison to the CALIPSO/CALIOP V4-20 cloud layer products and MODIS Collection 6.1 cloud top products. Each of these cloud properties show improvement with the use of these constructed channel radiances. The major improvement for the cloud mask is found over polar regions, where the correct cloud detection percentage increases due to decrease in missed cloud and/or false detection.



For cloud thermodynamic phase, the ice cloud fraction increases over non-polar regions and the combined liquid water and ice cloud discrimination improves in comparison with CALIPSO. The retrieved cloud top height for semitransparent ice clouds increases over non-polar regions and tends to be closer to the true CALIPSO/CALIOP cloud top height. Moreover, the uncertainty of

5 cloud top height retrievals decreases globally for these clouds.

# 1. Introduction

Current polar-orbiting satellite imager sensors, such as the Visible Infrared Imaging Radiometer Suite (VIIRS) onboard both the Suomi National Polar-orbiting Partnership (S-NPP) and the National Oceanic and Atmospheric Administration-20 (NOAA-20), have many advantages

10 compared to previous generation imagers, such as a wider scanning swath, a pixel size that varies little across the scan, and the addition of a day/night band (DNB). However, the absence of certain thermal infrared (IR) bands makes it challenging to accurately retrieve cloud properties that are dependent on those spectral measurements. For instance, VIIRS does not take measurements in the broad 6.7-µm water vapor band or the 15-µm $CO_2$ band that are useful for both cloud

15 thermodynamic phase and semitransparent ice cloud height retrievals (Baum et al. 2012). The 15-µm channels are used in the $CO_2$ slicing approach that was implemented in the National Aeronautics and Space Administration's (NASA) Moderate Resolution Imaging Spectroradiometer (MODIS) products (Menzel et al., 2008; Baum et al. 2012) and High resolution Infrared Radiometer Sounder (HIRS) products (Menzel et al. 2016). Fortunately, these IR channels

20 are available on the hyperspectral IR sensor called the Crosstrack Infrared Sounder (CrIS), also on the S-NPP and NOAA-20 platforms.

Previous studies retrieved cloud top height directly from sounder data (Susskind et al. 2003; Li et al. 2005; Kahn et al. 2007) but the spatial resolution of the sounder is much lower than that of the companion imager. Heidinger et al. (2019) developed a method to match sounder fields-of-view (FOVs; 15-km spatial resolution at nadir) and VIIRS imager pixels (750m at nadir), and adopted

the ice cloud top height retrieval from the sounder as the a-priori value to improve the imager-based cloud height retrieval using an optimal estimation approach. Here we denote field-of-view for the sounder and pixel for the imager exclusively to minimize confusion between the two sensors. This method has the advantage of using sounder information as an aid to retrieve products at imager resolution. There are three drawbacks to this approach: (1) both the imager and sounder

data need to be available during operational processing, (2) the algorithm must account for spatial gaps between sounder FOVs and the "stretching" of the FOVs towards the edge of the sounder scan swath, and (3) the sounder swath does not cover the entire imager swath.

To mitigate some of these limitations, this study employs an innovative data fusion approach

(Weisz et al. 2017) that constructs MODIS-like water vapor and $CO_2$ channel radiances directly at the imager resolution through use of VIIRS and CrIS radiances. To be clear, the data fusion method provides MODIS-like IR absorption channel radiances at the VIIRS M-band spatial resolution (750m). The VIIRS+CrIS fusion channel radiance products are available for the entire record of both S-NPP and NOAA-20 platforms, and access to these products is described in the Appendix.

The addition of these channels makes it possible to retrieve cloud properties, including cloud mask, cloud type/phase, and cloud top height products using algorithms developed and tested using the full MODIS channel suite. The goal of this study is to determine the impact of supplementing



VIIRS with the imager-resolution VIIRS-CrIS fusion channels on retrieving those three cloud products. This paper is organized as follows: Section 2 discusses data and retrieval methods; Section 3 presents results and findings, and a summary is provided in Section 4.

# 2. Data and Data Processing System

## 2.1 VIIRS Level-1 and Level-2 Data

The data used in this study include the standard Level–1B VIIRS data for both the S-NPP and NOAA-20 platforms made available by the Atmosphere Science Investigator-led Processing System (A-SIPS) located at the University of Wisconsin–Madison Space Science Engineering Center (SSEC). Only the M-band moderate spatial resolution (750m) VIIRS Level–1b data are used in this study.

The VIIRS+CrIS IR absorption channel radiances are available in a Level-2 product for the entire records of S-NPP and NOAA-20. A brief summary of the construction of high spatial resolution IR narrowband radiances is as follows. The method requires an accurate colocation between the high spatial resolution imager data (for VIIRS, M-band data are at 750m) and the lower-spatial-resolution sounder data (for CrIS, about 15 km). The fusion method consists of two steps: (a) performing a nearest neighbor search using a k-d tree algorithm on both high spatial (M-band data) and low spatial (M-band data averaged over the CrIS FOV) resolution split-window (11 and 12-μm) imager radiances, and (b) averaging the convolved sounder radiances at low spatial resolution for the five nearest neighbors selected in the previous step for each imager pixel. The term "convolved sounder radiances" refers to the process of applying a given spectral response function



(SRF) to the sounder hyperspectral radiances. The fusion product uses SRFs defined for the MODIS sensor on the NASA Earth Observation system (EOS) Aqua platform. Details on the data fusion methodology are in Weisz et al. (2017). The fusion products are available at the Level 1 and Atmosphere Archive and Distribution System (LAADS) Distributed Active Archive Center

(DAAC) at NASA Goddard Space Flight Center. The Appendix provides information related to documentation and access to this product.

## 2.2 CLAVR-x

The CLouds from AVHRR-Extended (CLAVR-x) processing system is used to retrieve cloud properties in this study. CLAVR-x is the operational processing system for the Advanced Very

High Resolution Radiometer (AVHRR) on NOAA's Polar Operational Environmental Satellites (POES) sensors. The Pathfinder Atmospheres Extended (PATMOS-x) is a climate dataset generated from CLAVR-x (Heidinger et al. 2014). Over time, CLAVR-x has become the development testbed for many of NOAA's operational cloud property retrieval algorithms using a variety of polar-orbiting and geostationary imagers, including the cloud mask (Heidinger et al.

2012), cloud top properties (Heidinger et al. 2019), Daytime Cloud Optical and Microphysical Properties (DCOMP; Walther and Heidinger 2012), cloud cover layer, and cloud base properties (Noh et al. 2017). Both daytime and nighttime cloud properties are retrieved within CLAVR-x except DCOMP, which uses reflectance channels only. CLAVR-x is available for public use and a    user    manual    is    available    from    the    following    website:

http://cimss.ssec.wisc.edu/clavrx/documentation.html.





The Cloud Mask retrieval algorithm is based on a Naive Bayesian approach (Heidinger et al. 2012). The algorithm uses a combination of visible (VIS: $0.4 < \lambda < 0.75$ μm), near-infrared (NIR: $0.75 < \lambda < 1.1$ μm), shortwave-infrared (SWIR: $1.1 < \lambda < 3$ μm), midwave-infrared (MWIR: $3 < \lambda < 5$ μm), and longwave-infrared (LWIR: $5 < \lambda < 15$ μm) channels to compute cloud probability based on a number of cloud tests for each pixel, and generates a 4-level cloud mask that classifies the pixel as cloudy, probably cloudy, probably clear and clear. In subsequent retrievals and validations, the pixel is considered cloudy if the 4-level mask shows cloudy or probably cloudy; otherwise, the pixel is declared clear. Cloud product retrievals are performed only on cloudy pixels.

10  The cloud type/phase retrieval is a critical part of the CLAVR-x system. It is based on a traditional decision tree method that uses measurements from the 1.6, 3.75, 8.5, 11 and 12-μm channels. If available, the 6.7 and 13.3-μm channels are also used for cloud thermodynamic phase retrievals, where they primarily impact the discrimination of semitransparent ice clouds.

15  Cloud top heights are retrieved with the GOES Algorithm Working Group (AWG) Cloud Height Algorithm (ACHA). ACHA employs an optimal estimation (OE) algorithm that uses LWIR channels only. ACHA derives the a-priori values based on cloud phase for its cloud top retrieval, so its performance relies on the phase algorithm. Also, ACHA does not process pixels sequentially. Instead, ACHA generates processing orders based on cloud phase, local radiative center (LRC), 20  and multilayer cloud detection. Here the use of LRC allows the algorithm to mitigate complexities arising from pixels having a very low cloud signal, such as cloud edges and optically thin ice clouds; it is defined as the pixel location, in the direction of the gradient vector of brightness temperature at 11 μm, upon which the gradient reverses or when a threshold value is found. Details

on ACHA can be found in its Algorithm Theoretical Basis Document (ATBD; also accessible from http://cimss.ssec.wisc.edu/clavrx/documentation.html). Cloudy pixels are assigned into different groups and processed based on group priority. Optically thin ice cloud pixels are processed in the last step using mean retrieved cloud top temperature from surrounding optically thicker ice cloud

pixels as the a-priori values.

Table 1 lists the channels used by the cloud mask, cloud type/phase and ACHA cloud top height algorithms, both with and without fusion channels. As shown in Table 1, the fusion water vapor channel at 6.7 μm can be used by both cloud mask and cloud type. The 13.3-μm channel is used

in the cloud type and ACHA modules but not in the cloud mask. The ACHA algorithm is versatile in that it supports various combinations of IR channels. Two combinations are tested in this study: one in which only the 13.3-μm fusion channel is added to the VIIRS 8.5, 11, and 12 μm channels, and one in which both the 6.7 and 13.3 μm channels are used in conjunction with the 8.5, 11, and 12 μm channels. It is difficult to explain definitively the information content available in each of

these IR bands so the approach is to test their impact on ice cloud height retrievals through comparison with another product. In this study, the comparisons are based on both the CALIPSO/CALIOP V4-20 cloud layer products and the MODIS Collection 6.1 cloud top products; these products are described in the following section.



| | No Fusion (µm) | Fusion (µm) |
|---|---|---|
| Cloud Mask | 0.41, 0.65, 0.87, 1.61, 2.25, 3.7, 8.5, 11, 12 | 0.41, 0.65, 0.87, 1.61, 2.25, 3.7, 8.5, 11, 12, 6.7 |
| Cloud Phase | 1.61, 3.7, 8.5, 11, 12 | 1.61, 3.7, 8.5, 11, 12, 6.7, 13.3 |
| ACHA | 8.5, 11, 12 | 8.5, 11, 12, 6.7*, 13.3 |

Table 1. Spectral channels used in the retrievals for fusion and no fusion experiments. The asterisk indicates that we present ACHA results with and without the 6.7-µm channel (results are without 6.7-µm unless inclusion is noted).

## 2.3 Comparison Datasets

The Cloud-Aerosol Lidar with Orthogonal Polarization (CALIOP) instrument is a near-nadir viewing lidar system onboard the Cloud-Aerosol Lidar and Infrared Pathfinder Satellite Observations (CALIPSO; Winker et al. 2009). CALIOP sends active lidar signals downward which can penetrate the atmospheric layers and provide vertical profiles of clouds and aerosols. CALIPSO was part of NASA's A-Train constellation from 2006 until 2018, when it left the A-

train for a lower orbit to stay in sync with CloudSat. However, it continues to provide reliable global observations. The orbits of CALIPSO/CALIOP overlap with both S-NPP and NOAA-20 periodically. Over time, the orbits coincide enough to provide global coverage. The CALIPSO products offer a unique assessment of VIIRS cloud retrievals.

In this study, collocations with CALIPSO/CALIOP are studied for two weeks of S-NPP data from April and October 2018 and one week of NOAA-20 data from January 2019. Collocations with

CALIPSO are selected as described in Heidinger et al. (2019). Briefly summarized, the time difference must be within 15 minutes between VIIRS and CALIPSO and the spatial distance must be within 4°. To make use of the full potential of CALIPSO/CALIOP data, the 1-km and 5-km products are combined when clouds are not reported in the 5-km product. While both Version 3

and 4 CALIPSO/CALIOP products are available, the latest Version 4-20 cloud layer product is used (Vaughan et al., 2018; Avery et al., 2019). In this paper, the true CALIPSO/CALIOP cloud top height for the uppermost cloud layer is used for validation instead of an adjusted CALIPSO/CALIOP cloud top height as described in Heidinger et al. (2019).

The Aqua-MODIS Collection 6.1(C6.1) cloud height products are used as an additional comparison dataset. Cloud top heights in C6.1 are retrieved with the $CO_2$ slicing technique that uses a combination of $CO_2$ absorption bands (Menzel et al. 2008). Key features of cloud top property refinements for Collection 6 are described in Baum et al. (2012). The collocation tools developed by the Atmosphere SIPS are used to generate collocations between S-NPP/NOAA-20

and Aqua.

# 3. Results

## 3.1 Cloud Mask

Cloud mask retrievals are compared to collocated CALIPSO/CALIOP, with results presented in Table 2. In assessing the cloud mask product, the CALIPSO/CALIOP cloud fraction is used to

classify the pixel as to its cloud/clear state. A cloud fraction of 1 means it is cloudy, and a fraction of 0 implies that the pixel is clear. Pixels with values in between are discarded to avoid cloud edges



and the potential for partially-cloudy pixels. Additionally, pixels with CALIPSO/CALIOP cloud optical depth lower than 0.03 are filtered to exclude sub-visible clouds from the perspective of VIIRS. Table 2 shows the sample sizes and percentages of correct, missed and false detected clouds for different geographical regions for S-NPP and NOAA-20, where a correct detection

means that the pixel is classified as cloudy by both VIIRS and CALIPSO/CALIOP. If VIIRS reports clear and CALIOP indicates cloudy, it is classified as a missed cloud. If VIIRS reports cloudy and CALIOP reports clear, the classification is regarded as a false cloud.

From a global perspective, adding the fusion channels tends to increase the correct overall detection percentage and decrease both missed and false cloud percentages. This applies to both

platforms and the impact appears to be slightly better for NOAA-20 (from 81.7% to 82.8%) than S-NPP (from 82.5% to 83.3%). A regional analysis indicates that the increase in correct detections occurs primarily over polar regions. The most pronounced change is over the Arctic in the NOAA-20 product, which shows that correct detection increases from 61.9% to 67.6% and the false detection decreases from 13.7% to 7.4%. Unlike S-NPP, results of NOAA-20 do not always show

improvement in missed cloud and false detection, which is likely due to differences in orbits, observation geometry, sensor characteristics, etc. Cloud detection over snow-covered surfaces is a challenging problem, and the overall increase of correct detection clearly demonstrates the positive impact of the fusion channels. Over nonpolar regions, a slight increase of correct detection of 0.2 is seen for both platforms. This is unsurprising since the cloud mask algorithm performs

fairly well for a snow-free surface even without the water vapor channel. The general conclusion does not change when the optical thickness threshold is changed; the improvement in cloud detection is always observed. This indicates that inclusion of the fusion channel is valuable for cloud detection in problematic regions, without causing negative impacts in other regions.



| | | Sample Size | | Correct Detection | Missed Cloud | False Detection |
|---|---|---|---|---|---|---|
| S-NPP | Global | 5873247 | Fusion | 83.3 | 12.5 | 4.2 |
| | | | No Fusion | 82.5 | 12.8 | 4.7 |
| | 60°N to 60°S | 4207459 | Fusion | 85.8 | 10.7 | 3.5 |
| | | | No Fusion | 85.6 | 10.8 | 3.6 |
| | Arctic | 836038 | Fusion | 76.9 | 15.4 | 7.6 |
| | | | No Fusion | 74.7 | 16.8 | 8.4 |
| | Antarctic | 829750 | Fusion | 77.2 | 18.5 | 4.3 |
| | | | No Fusion | 74.7 | 19.1 | 6.2 |
| NOAA-20 | Global | 2254727 | Fusion | 82.8 | 13.1 | 4.1 |
| | | | No Fusion | 81.7 | 13.3 | 5.0 |
| | 60°N to 60°S | 1586709 | Fusion | 85.5 | 10.7 | 3.8 |
| | | | No Fusion | 85.3 | 10.8 | 3.9 |
| | Arctic | 319328 | Fusion | 67.6 | 25.0 | 7.4 |
| | | | No Fusion | 61.9 | 24.4 | 13.7 |
| | Antarctic | 348690 | Fusion | 84.4 | 12.9 | 2.7 |
| | | | No Fusion | 83.3 | 14.4 | 2.3 |

Table 2. Validation of cloud mask detection against CALIPSO/CALIOP using data collocated globally. Data with cloud optical depth less than 0.03 are filter.

Figure 1 shows the global cloud fraction averaged over the study period. Consistent with Table 2, the difference plots show that false cloud detection exists in polar regions in S-NPP in both hemispheres. There is also substantial false cloud detection in the NOAA-20 products over the Arctic. Additionally, missed clouds (VIIRS reports clear and CALIOP indicates cloudy) are

5    prevalent over the Antarctic in the NOAA-20 product, as shown in Table 2.

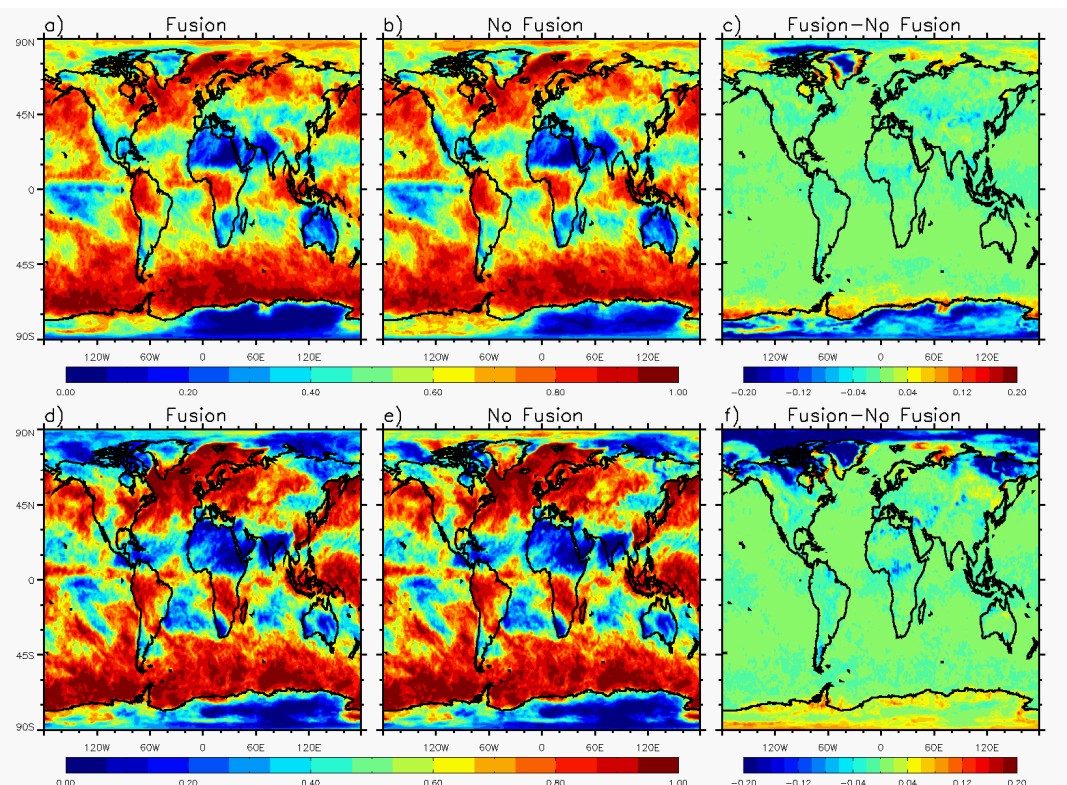

Figure 1. Mean gridded cloud fraction with fusion channels (left column), without fusion channels (middle column) and differences between fusion and no fusion (right column). The upper row shows S-NPP and lower row shows NOAA-20.

10   ## 3.2 Cloud Phase

Misidentification of the cloud phase (i.e., retrieving liquid water clouds as ice phase, and vice versa) directly affects ACHA as it relies on accurate cloud phase discrimination. CLAVR-x uses both the 6.7µm and 13.3µm channels, if available, in its cloud type retrieval algorithm. Table 3



demonstrates the impact of the fusion channels by comparing VIIRS-retrieved cloud phase to CALIPSO/CALIOP. The percentages of both correctly identified and incorrectly identified cloud phase pixels are shown. The total of all four categories adds up to 100%. The percentage of correct identifications for the ice category increases by about 2% for both S-NPP and NOAA-20 when 5 fusion channels are included. However, it also reveals that adding fusion channels tends to slightly decrease the correct identification of liquid water cloud pixels by about 1.5%.

| | | | CALIPSO/CALIOP | |
| --- | --- | --- | --- | --- |
| | | | Ice | Water |
| S-NPP | Fusion | Ice | 39.4 | 4.5 |
| | | Water | 19.3 | 36.8 |
| | No Fusion | Ice | 36.0 | 3.6 |
| | | Water | 22.2 | 38.2 |
| NOAA-20 | Fusion | Ice | 39.8 | 5.3 |
| | | Water | 15.7 | 39.2 |
| | No Fusion | Ice | 36.9 | 4.3 |
| | | Water | 18.1 | 40.7 |

Table 3. Percentages (%) of global cloud phase detection when a valid CTH retrieval is available comparing CALIPSO/CALIOP and CLAVR-x S-NPP VIIRS under both fusion and no-fusion 10 cases.

A geographical distribution plot similar to Figure 1, but for ice cloud only, is shown in Figure 2. The difference plots for both platforms are generally consistent with that in Figure 1. This is





because polar clouds are mostly ice clouds, so changes in total cloud fraction over the polar regions are also seen in the ice cloud fraction. This is confirmed by examining the water cloud fractions which show relatively subtle changes from the addition of the fusion channels (not shown). An increase in ice cloud fraction other than in polar regions is observed, though changes in total cloud

5   fraction are subtle.

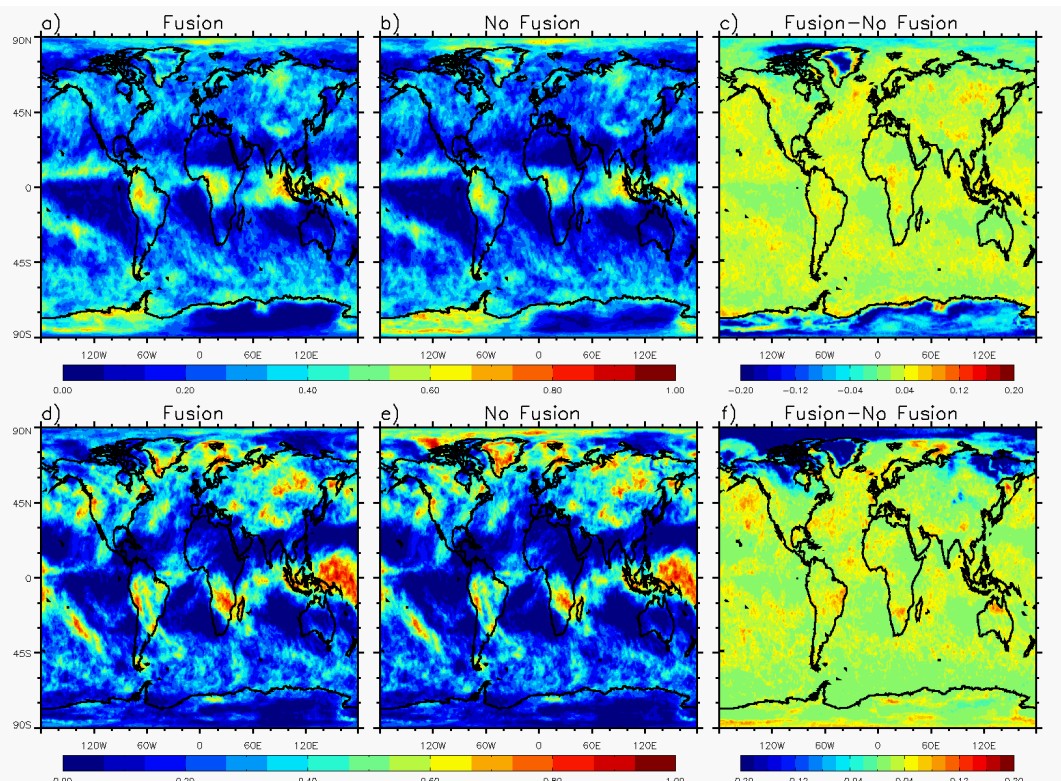

Figure 2. Mean gridded ice cloud fraction with fusion channels (left column), without fusion channels (middle column) and differences between fusion and no fusion (right column). The upper row shows S-NPP and lower row shows NOAA-20.

10   ## 3.3 Cloud Top Height

In the assessment of cloud top height, similar analyses are conducted as in Heidinger et al. (2019). However, as noted earlier, one major difference in this study is the use of the true CALIPSO/CALIOP cloud top instead of the adjusted value. The IR cloud top retrieval inevitably





is lower compared to the lidar height. Figure 3 shows an image of 11μm brightness temperatures from S-NPP VIIRS and cloud top height retrievals from MODIS C6.1, and S-NPP VIIRS with and without fusion over the tropical Pacific. Only results for ice clouds are shown. Compared to MODIS, semitransparent ice cloud heights are significantly underestimated using VIIRS channels

5      only (Figure 3c). Figure 3d shows that using the additional information provided by the fusion 13.3um channel improves the retrieval and brings results closer to MODIS C6.1. This clearly shows that ACHA's optimal estimation approach benefits from the fusion channel information.

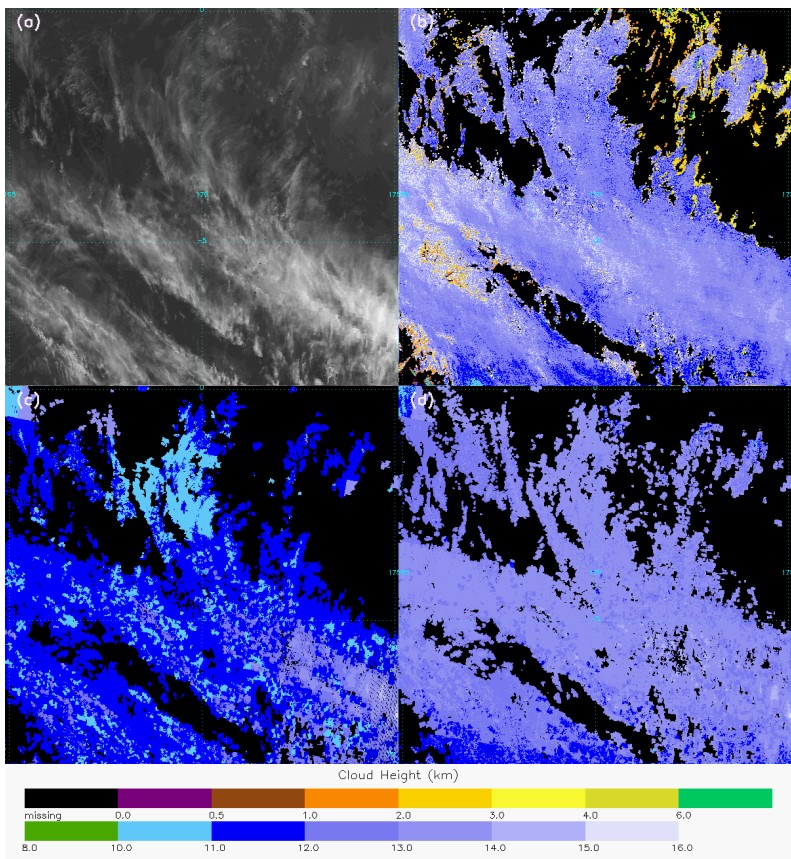

Figure 3. A cirrus cloud scene over tropical Pacific showing a) brightness temperature from S-
10    NPP VIIRS, b) MODIS C6.1, c) no fusion VIIRS, and d) fusion cloud top height for ice clouds only.



Figure 4 plots the histogram of cloud top height bias of ice phase clouds in comparison to CALIPSO/CALIOP for different cloud emissivity ranges for S-NPP VIIRS. Only single layer clouds as reported by CALIPSO/CALIOP are included and both cloud phase and emissivity are matched for each product. As expected, the passive IR-based cloud top height retrieval is lower

than in the lidar product. The largest bias is seen for the group with the lowest emissivity. Cloud heights based solely on the VIIRS IR window channels shows that there is a significant fraction of ice clouds that shows negative biases greater than 4km in the two groups with smaller emissivity ranges (Figure 4a and 4b). The retrievals improve significantly if the fusion 13.3-µm channel is used. For optically thicker ice clouds (emissivity between 0.8 to 1.0), the performances from both

retrievals are similar as window channels do fairly well for optically thick clouds. In general, when all ice clouds are considered (Figure 4d), the improvement is still quite apparent. In Figure 5, the zonal means of the cloud top height biases are plotted for different emissivity ranges. The noticeable feature is a dramatic improvement over tropical regions when the emissivity is less than 0.8 (Figs. 5a and 5b), where semitransparent ice clouds are most abundant and the underestimation

occurs the most frequently. The impact for high latitude regions is generally negligible.



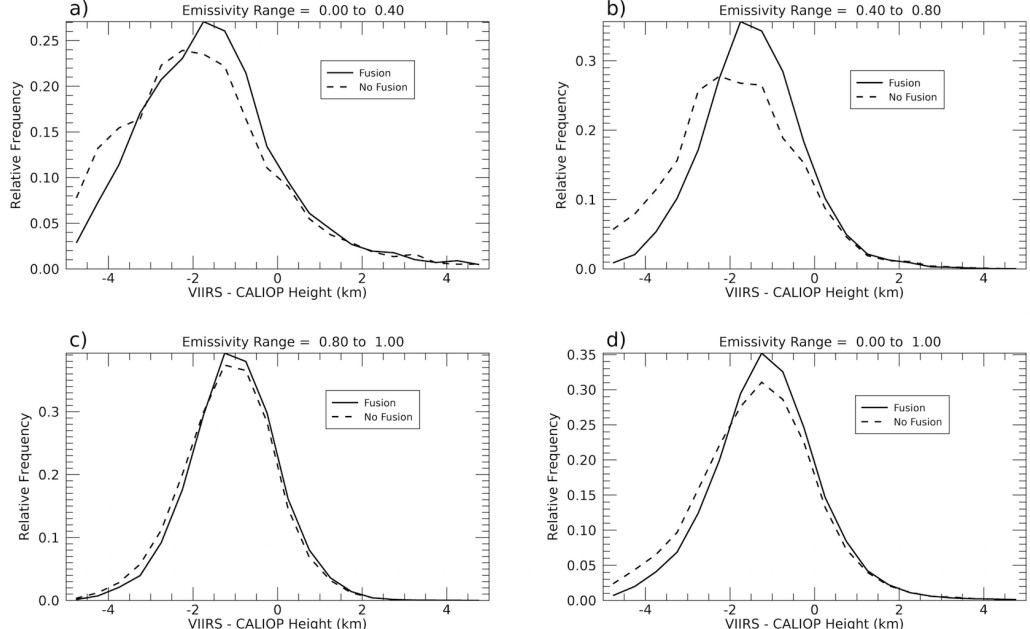

Figure 4. Bias distribution of cloud top height between S-NPP VIIRS and CALIPSO/CALIOP for emissivity range a) 0 to 0.4; b) 0.4 to 0.8; c) 0.8 to 1.0; and d) 0 to 1.0. Solid and dashed lines indicate data with/without fusion channels.




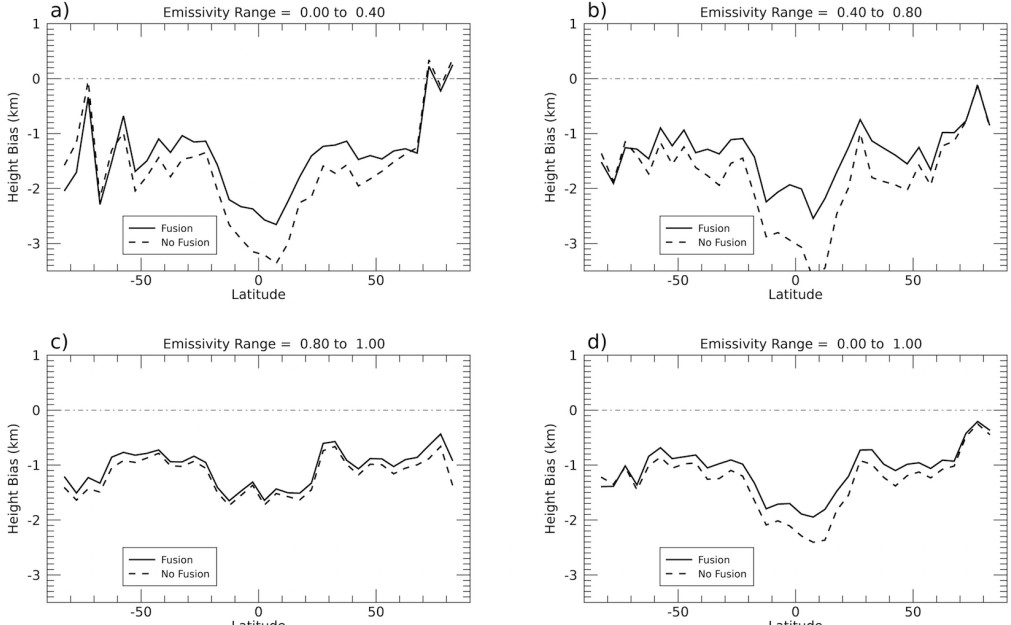

Figure 5. Zonal distribution of cloud height biases between S-NPP VIIRS and CALIPSO/CALIOP for emissivity range a) 0 to 0.4; b) 0.4 to 0.8; c) 0.8 to 1.0; and d) 0 to 1.0. Solid and dashed lines indicate data with/without fusion channels.

Table 4 presents the mean, standard deviation and mode of biases as well as counts of pixels. Not

10    only do the mean biases improve in all cases, but also the standard deviation decreases uniformly.

The modes also tend to be closer to 0 except for thick clouds.



Atmospheric
Measurement
Techniques

Discussions

| Emissivity | | Counts | Bias (km) | Standard Deviation (km) | Mode (km) |
|---|---|---|---|---|---|
| 0 to 0.4 | No fusion | 62941 | -1.96 | 2.07 | -2.25 |
| | Fusion | | -1.62 | 1.86 | -1.75 |
| 0.4 to 0.8 | No fusion | 22190 | -1.95 | 1.54 | -2.25 |
| | Fusion | | -1.46 | 1.23 | -1.75 |
| 0.8 to 1.0 | No fusion | 227330 | -1.15 | 1.10 | -1.25 |
| | Fusion | | -1.04 | 1.06 | -1.25 |

Table 4. Statistics of differences between S-NPP VIIRS cloud top height and CALIPSO-CALIOP using two weeks of data in April and October in 2018, when fusion data are used/unused for four emissivity ranges. Emissivity from both ACHA and derived from CALIPSO/CALIOP cloud optical depth are applied.

Similar analyses are also performed on NOAA-20 cloud top height products in Figure 6, Figure 7 and Table 5. It is observed that though the counts in Table 5 are smaller than for S-NPP (since only one rather than two weeks was processed for NOAA-20), positive impacts on cloud top

10    heights are revealed and the performance is consistent between S-NPP and NOAA-20.

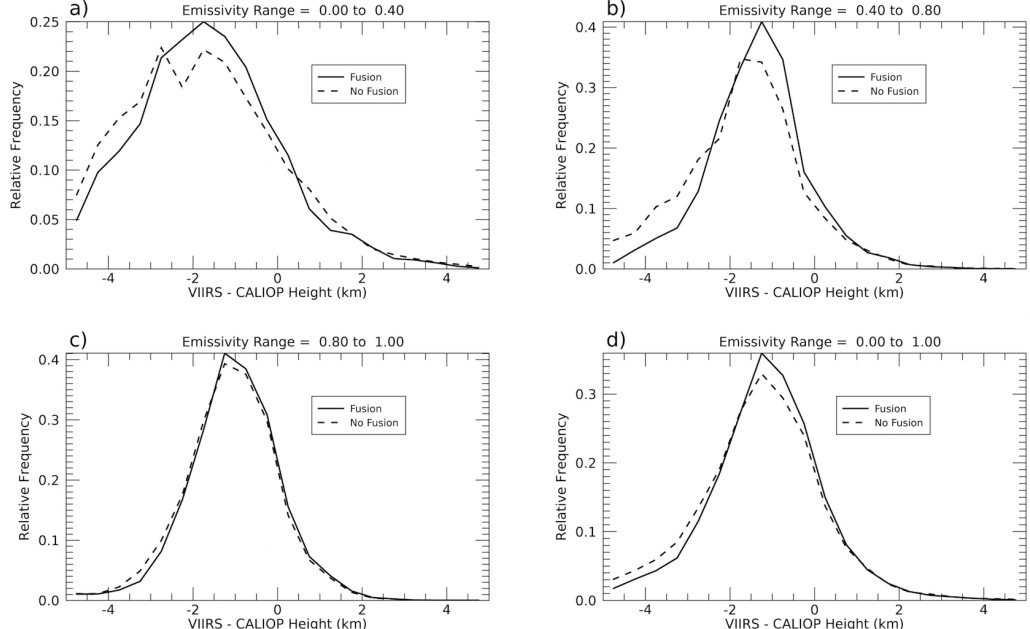

Figure 6. Bias distribution of cloud top height between NOAA-20 VIIRS and CALIPSO/CALIOP for emissivity range a) 0 to 0.4; b) 0.4 to 0.8; c) 0.8 to 1.0; and d) 0 to 1.0. Solid and dashed lines indicate data with/without fusion channels.





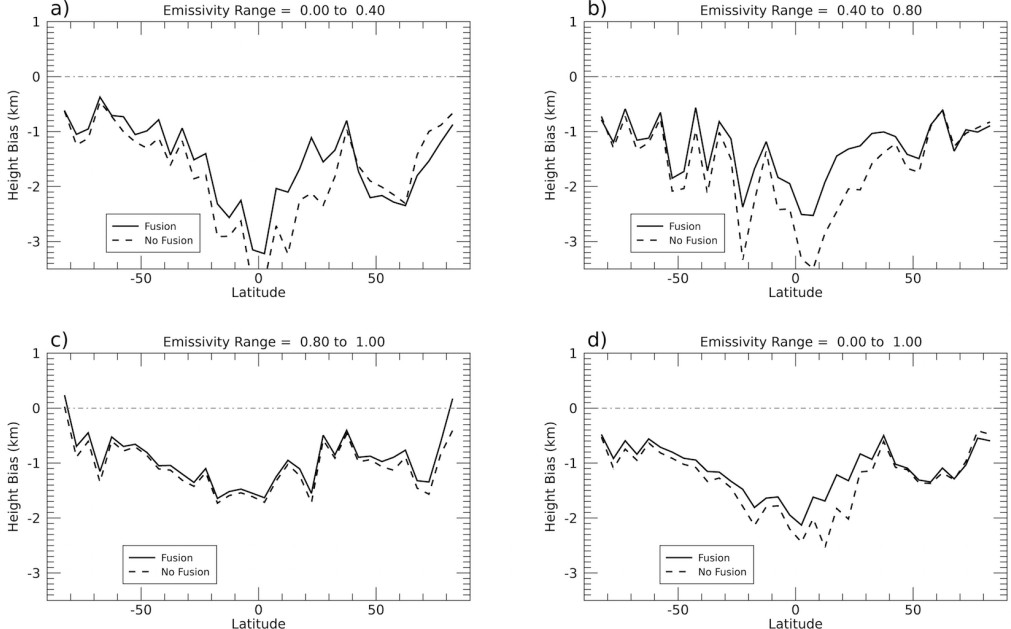

Figure 7. Zonal distribution of cloud height biases between NOAA-20 VIIRS and CALIPSO/CALIOP for emissivity range a) 0 to 0.4; b) 0.4 to 0.8; c) 0.8 to 1.0; and d) 0 to 1.0. Solid and dashed lines indicate data with/without fusion channels.



| Emissivity | | Counts | Bias (km) | Standard Deviation (km) | Mode (km) |
|---|---|---|---|---|---|
| 0 to 0.4 | No fusion | 28875 | -2.02 | 2.07 | -2.75 |
| | Fusion | | -1.83 | 1.85 | -1.75 |
| 0.4 to 0.8 | No fusion | 7192 | -1.78 | 1.51 | -1.75 |
| | Fusion | | -1.37 | 1.19 | -1.25 |
| 0.8 to 1.0 | No fusion | 85079 | -1.12 | 1.12 | -1.25 |
| | Fusion | | -1.03 | 1.09 | -1.25 |

Table 5. Statistics of differences between NOAA-20 VIIRS cloud top retrieval and CALIPSO/CALIOP using one week of data in January 2019, when fusion data are used/unused for three emissivity ranges.

To demonstrate the impact of fusion water vapor channel on cloud height retrievals, the zonal means of S-NPP cloud top height biases retrieved with both 6.7-μm and 13.3-μm compared to VIIRS-only channels are displayed in Figure 8. Compared to adding only the 13.3-μm fusion channel, cloud heights tend to increase and match more closely to those from CALIPSO/CALIOP. This is observed not only for optically thin clouds with emissivities less than 0.4 but also for clouds in the 0.4 to 0.8 emissivity range. Therefore, the water vapor channel adds to the information available from the 13.3-μm $CO_2$ band. The optimal use of the fusion channels deserves further study.



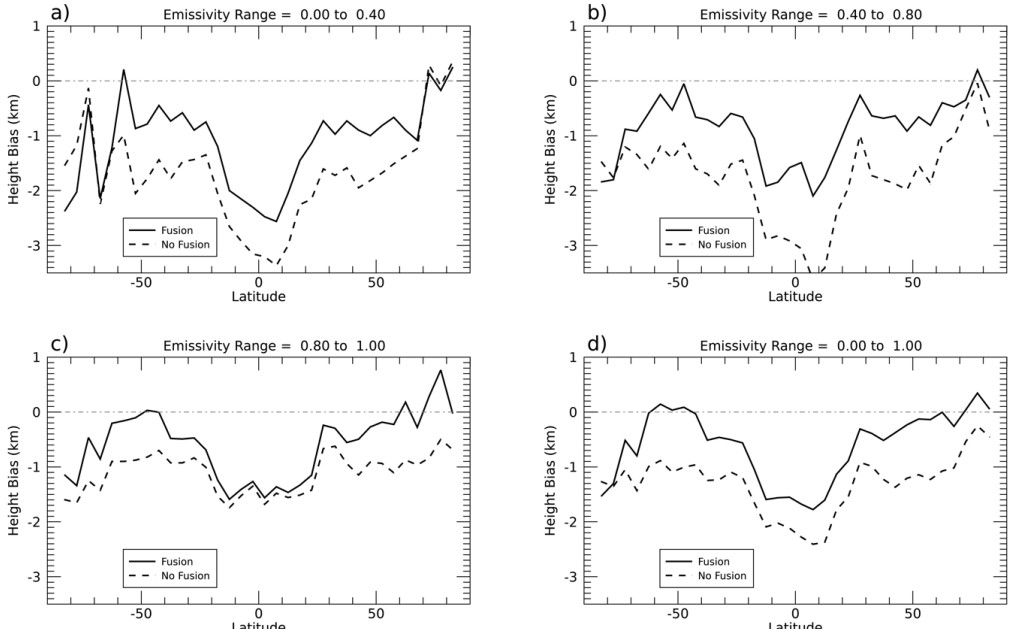

Figure 8. Zonal distribution of cloud height biases between S-NPP VIIRS and CALIPSO/CALIOP for emissivity range a) 0 to 0.4; b) 0.4 to 0.8; c) 0.8 to 1.0; and d) 0 to 1.0. Solid and dashed lines indicate data with/without fusion channels. Both 6.7-µm and 13.3-µm fusion channels are used for the fusion experiment.

While both Heidinger et al. (2019) and this study address the same problem with different approaches involving similar channels from the CrIS sounder, both studies show positive impact when using the sounder channels. However, Heidinger et al. (2019) showed that the major improvement of ice cloud height retrieval was for those in emissivity ranges 0 to 0.4. This study suggests that using the fusion channels may have a greater impact on the ice clouds with emissivity ranges between 0.4 and 0.8.



## 3.4 Cloud Top Height Retrieval Uncertainty

Estimation of retrieval uncertainty is an important output from the optimal estimation approach. The retrieval uncertainty measures the confidence of the retrieval product. A lower uncertainty can be interpreted as there being a higher confidence in the retrieval results, and vice versa. In the

optimal estimation output, the retrieval uncertainty is the square root of the diagonal component of the error covariance matrix of the retrievals. ACHA first generates retrieval an uncertainty for each of the retrieved parameters including cloud top temperature, and the uncertainty of cloud top height is derived subsequently by dividing the cloud top temperature uncertainty by a lapse rate. Here a constant lapse rate of 7K/km is used. Figure 9 shows the zonal mean retrieval uncertainty

of the ice cloud top heights using global sub-sampled Level 2b data, which are derived at a 0.1˚ by 0.1˚ spatial resolution using a nearest neighbor nadir-overlap sampling technique. Level 2b subsampled data are computed daily from level 2 data separately for ascending and descending tracks. Several features are noticed: 1) the uncertainties are smaller with variations between 1.0km and 1.5km between 60°N and 60°S; 2) the uncertainties increase gradually poleward of 60° and

the maximum values are about 0.2km (~2K) at both hemispheres; 3) results using fusion channels reduce uncertainties across all latitudes and the major improvement is between 60°N and 60°S.



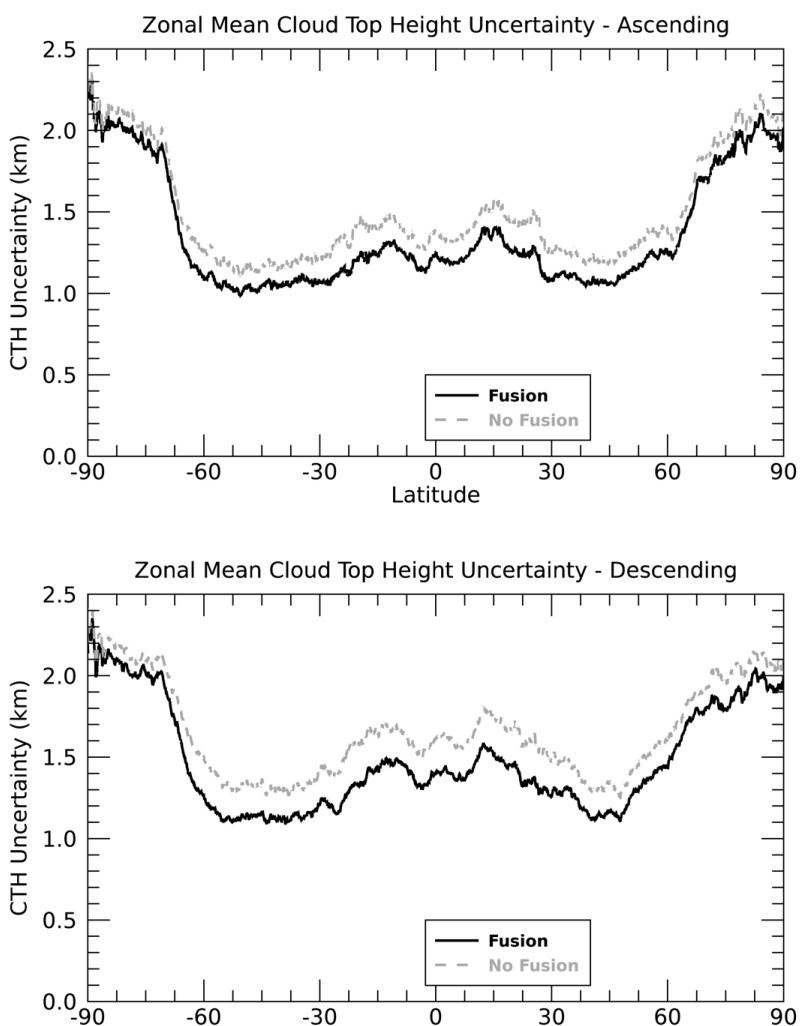

Figure 9. Zonal mean ice cloud top height uncertainty estimated from ACHA's optimal estimation algorithm for S-NPP VIIRS computed from two weeks of global Level 2b data in 2018. The top panel shows the ascending track and the bottom panel shows the descending track.

Figure 10 shows ice cloud top height uncertainty as a function of cloud emissivity, derived from

the same global Level 2b data as in Figure 9. Larger differences for ice clouds with smaller

emissivities are expected and this result is supported by the results in Figure 10. As emissivity

increases above 0.8, the differences tend to decrease gradually. It is also observed that the


differences are negligible when emissivity is less than 0.05, which can be explained by the

limitation of passive sensors such as VIIRS in detecting such optically thin clouds.

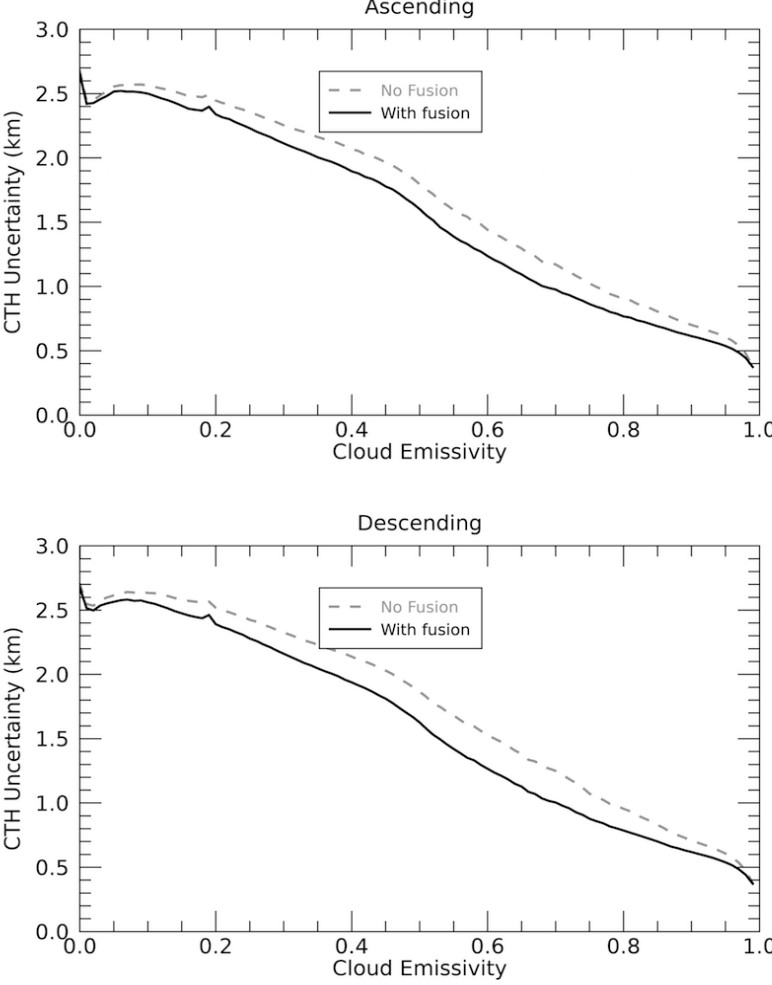

Figure 10. Ice cloud top height uncertainty as a function of cloud emissivity for the S-NPP VIIRS
5   computed from two weeks of global Level 2b data in 2018.



# 4. Summary and Discussion

The absence of water vapor and $CO_2$ absorption IR channels on the VIIRS imager on the Suomi-NPP and NOAA-20 polar-orbiting platforms limits the capability for cloud property retrievals, especially for retrievals involving semitransparent ice clouds. This study shows the advantage of

using two IR absorption channels at 6.7 and 13.3 μm that are constructed at VIIRS spatial resolution (750m) using a data fusion approach using both sounder (CrIS) and imager (VIIRS) measurements following Weisz et al. (2017). The positive impact of using the constructed 6.7 and 13.3-μm fusion channels on three commonly retrieved cloud properties (cloud mask, type/phase, and cloud top height) is demonstrated. The cloud retrievals are based on the NOAA operational

CLouds from AVHRR-extended (CLAVR-x) retrieval package. The cloud height module is called the AWG Cloud Height Algorithm (ACHA), where AWG refers to the Algorithm Working Group set up a number of years ago in preparation for working with data from the GOES Advanced Baseline Imager. Evaluation of the resulting cloud products are performed through comparison to the CALIPSO/CALIOP V4-20 cloud layer products and MODIS Collection 6.1 cloud top products.

We note that improvements are observed for all three products when quantitatively compared to the CALIPSO/CALIOP products. Each of these cloud properties show improvement with the use of the 6.7 and 13.3-μm fusion channel radiances. The major improvement for cloud mask is over polar regions, where percentage of cloud detection increases due to decrease in missed cloud

and/or false cloud detection.

With regard to cloud thermodynamic phase, the ice cloud fraction increases over non-polar regions and the combined detection rates for both water and ice clouds also increase. The impact of using

IR absorption channels in this study are similar to the impact shown in MODIS Collection 6 products that added similar channels to improve the approach in Collection 5 that used only the 8.5, 11, and 12-µm IR window channels (Baum et al. 2012).

The retrieved cloud top height for semitransparent ice clouds increases in non-polar regions and tends to be closer to the true CALIPSO/CALIOP cloud top. The retrievals obtained using the 13.3-µm channel in addition to the 8.5, 11, and 12-µm IR window channels are improved over those obtained solely with the IR window channels. The retrieved semitransparent ice cloud heights are closer to the CALIPSO V4-20 product, and both the biases and standard deviations decrease. The

inclusion of a channel at 6.7-µm further decreases the bias and standard deviation values. This suggests that there is room for additional improvement in the cloud height retrievals by testing different combinations of the IR absorption fusion channels. The positive impact on cloud heights, as compared to CALIPSO, is seen at all latitudes for both Suomi-NPP and NOAA-20 platforms, and the uncertainty in the cloud top height retrievals decreases at almost all latitudes.

The approach described in Heidinger et al. (2019) also used a combination of VIIRS and CrIS radiance data to demonstrate the potential for improving ice cloud retrievals. With the data fusion product available for VIIRS, however, the constructed IR absorption channel radiances are provided at VIIRS M-band (750m) spatial resolution for the full imager swath. The fusion results

indicate a positive impact in cloud height over a range in emissivity up to 0.8. The results in this study are limited to a VIIRS sensor scan angle of 50˚ to minimize the impact of the sounder swath being less than that of the imager. These findings are limited in scope but clearly demonstrate the





potential in the use of the fusion IR absorption channels in generating cloud products. In future work, we plan to extend this evaluation to longer time periods.

**Data availability**. The VIIRS Level-1 data and Level-2 fusion products used in this study were
obtained from the A-SIPS data archive (https://sips.ssec.wisc.edu/#/products/list, last access: December 26, 2019). Currently the VIIRS Level-1 and Level-2 fusion data are accessible to the public, free of charge, from the LAADS data center, and more information is provided in the Appendix. The following CALIPSO standard data products were used in this study: the CALIPSO Level-2 1-km cloud layer product V4-20 (Vaughan et al., 2018; NASA Langley Research Center
Atmospheric                Science                Data                Center; https://eosweb.larc.nasa.gov/project/calipso/cal_lid_l2_01kmclay-standard-v4-20, last access: December 26, 2019); the CALIPSO Level-2 5-km cloud layer product V4-20 (Vaughan et al., 2018; NASA Langley Research Center Atmospheric Science Data Center; https://eosweb.larc.nasa.gov/project/calipso/cal_lid_l2_05kmclay_standard_v4_20, last access:
December 26, 2019). MODIS data comparisons were conducted using the MODIS Collection 6.1 Atmosphere L2 MYD06 Cloud Product (https://ladsweb.modaps.eosdis.nasa.gov/missions-and-measurements/products/MYD06_L2, last access: December 26, 2019)

**Competing Interests**. The authors declare that they have no conflict of interest.

**Author Contributions**. Yue Li conducted the impact study of using fusion channels, performed the analyses and prepared the figures as well as the first draft of the manuscript. Bryan A. Baum and W. Paul Menzel made critical suggestions on the design of the study and significant





improvements to the manuscript. Andrew K. Heidinger gave guidance on the use of CLAVR-x and interpretation of results. Elisabeth Weisz provided expertise on the use of fusion products.

**Acknowledgements**. The authors gratefully acknowledge support from NASA grant
80NSSC18K0816 and the encouragement of Dr. Hal Maring (NASA Headquarters, Washington, DC). Yue Li and Andrew K. Heidinger also acknowledge the support from NOAA grant NA15NES4320001. The fusion data are generated by the Atmosphere SIPS at University of Wisconsin – Madison and are available at LAADS (see Appendix). The writing of this paper benefited from discussions with our colleague Denis Botambekov. We thank Geoff Cureton for
his effort at the A-SIPS and Bhaskar Ramachandran for help in staging the fusion product at LAADS and providing link to the data.

# Appendix: Accessing the VIIRS+CrIS Fusion Products

The Level-1 and Atmosphere Archive & Distribution System (LAADS) data center manages and
hosts VIIRS+CrIS fusion products derived from the Suomi National Polar-orbiting Partnership (S-NPP) and NOAA-20 platforms. The following links provide access to users interested in acquiring these products, which are free of charge. All users need to register with NASA Earthdata to obtain a login account through the NASA User Registration System (URS) page (https://urs.earthdata.nasa.gov). For additional help on any aspect of searching for or acquiring
these products, contact the LAADS User Services: http://MODAPSUSO@lists.nasa.gov.

1. **The VIIRS+CrIS Fusion product page (provides overview and documentation):**
   https://ladsweb.modaps.eosdis.nasa.gov/missions-and-measurements/science-domain/viirs-cris-fusion/
2. **Perform a specific geographical search for the S-NPP VIIRS+CrIS fusion product:**
   https://ladsweb.modaps.eosdis.nasa.gov/search/order/2/FSNRAD_L2_VIIRS_CRIS_SNPP—5110



3. **Perform a specific geographical search for the NOAA-20 VIIRS+CrIS fusion product:**

   https://ladsweb.modaps.eosdis.nasa.gov/search/order/2/FSNRAD_L2_VIIRS_CRIS_NOAA20--5110

4. **Direct access to the S-NPP VIIRS+CrIS fusion product archive:**

   https://ladsweb.modaps.eosdis.nasa.gov/archive/allData/5110/FSNRAD_L2_VIIRS_CRIS_SNPP/

5. **Direct access to the NOAA-20 VIIRS+CrIS fusion product archive:**

   https://ladsweb.modaps.eosdis.nasa.gov/archive/allData/5110/FSNRAD_L2_VIIRS_CRIS_NOAA20

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
