# Peer review of "Improvement in cloud retrievals from VIIRS through the use of infrared absorption channels constructed from VIIRS-CrIS data fusion"

_Atmospheric Measurement Techniques, 2019_

## Referee Comment (RC1) · Anonymous Referee #1 · 25 Mar 2020

The present manuscript describes and validates the improvement of cloud retrievals from the VIIRS instrument on board of Suomi-NPP platform using radiances from CrIS hyperspectral instrument on board of the same platform.

The authors, using a fusion methodology, extracted broadband channels information from CrIS spectrally resolved measurements for simulate MODIS channels around 15 micron and 6.7 micron. In this way they can apply methodologies developed for MODIS to VIIRS that doesn't cover these spectral bands for cloud detection and retrieval. This improvement is been validated with CALIPSO dataset.

[Figure]

The manuscript topic is for sure appropriate for the Journal but in the present form has some incompleteness that should to be fit before publication. Incompleteness can be identified divided into two main topics: Hyperspectal instruments and Validation.

- Regarding hypespectral instruments as I said, in this work the authors use the spectrally resolved measurements of CrIS to simulate moderate resolution channels. In doing this the authors omitted to describe and acknowledge the great diagnostic power inside the spectral resolution and coverage of instruments like CrIS. For a reader who is not an expert in the field, it might appear that CrIS (and all the hyperspectral instruments) is a less accurate instrument than VIIRS because it has a worse spatial resolution. As an example, consider sentence at lines 15-21 of page 2 and lines 1-5 of page 3. It seems that CrIS has channels at 15 and 6.7 micron, missing in VIIRS instrument, but with degraded spatial resolution. I think that the authors should spend a sentence to indicate the peculiarities of hyperspectral instruments and add a figure showing a typical CrIS measurement in comparison with the spectral coverage of the channels used in the methodology described in the manuscript. Moreover I wish to recall that already 15 years ago it has been shown that with hyperspectral observation alone in the atmospheric window between 800-900 cm$^{-1}$ is possible to detect and classify clouds. The authors can find an example in the following papers doi:10.1364/AO.41.000965 and doi:10.1016/S0022-4073(02)00083-3.

- About the validation, I have some doubts regarding the spatial distance between VIIRS and CALIPSO used for the colocation. While on the one hand I can imagine that a distance of less than 4° can reduce the concomitances between the two instruments, on the other a distance of 200 kilometers make the difference in spatial resolution between VIIRS and CrIS practically not appreciable. Probably a sentence that best justifies this choice is necessary. Also in relation to the results of the validation itself.

Other Specific Points

- Page 2. Line 19. As I said before, CrIS has not only channels MODIS-like at 6.7 and 15 microns, but it covers the spectral ranges that MODIS cover with two channels with thousand channels.

- Page 4. Line 11. Remove absorption before channel.

- Page 4. Line 19. The step (b) of the fusion method is not clear. The convolved sounder radiances are already at coarser spatial resolution. In the text it seems that the authors further degraded spatial resolution. Please clarify.

- Page 6. Line 16. Please insert a reference to the ACHA algorithm. If not, please place here the reference to the ATBD now at Page 7, line 2)

For these reasons I suggest to accept this manuscript subject to minor but necessary revisions.

---

## Referee Comment (RC2) · Anonymous Referee #2 · 17 Apr 2020

General comments:

Retrieval of cloud-top properties with the Visible Infrared Imaging Radiometer Suite (VIIRS) could be more challenging than its predecessor MODIS, because of the lack of water vapor and CO2 bands in thermal infrared region. This paper "Improvement in cloud retrievals from VIIRS through the use of infrared absorption channels constructed from VIIRS-CrIS data fusion" by Li et al. demonstrated that by leveraging fusion water vapor and CO2 bands from high-spectral resolution instrument CrIS, VIIRS cloud retrievals, including cloud mask, cloud thermodynamic phase, and cloud-top height are generally improved. This paper also shows that those fusion bands have a big boost in
the accuracies of cloud mask/phase algorithm at high latitude. By including the extra fusion bands, cloud-top height retrieval is also improved with lower biases and uncertainties, in particular for those optically thin cirrus clouds with emissivity less than 0.8.

This paper is well organized and written. One of my major concern is that the authors should give more details about the comparisons between VIIRS retrievals and CALIPSO/CALIOP. Furthermore, to highlight the importance of those absorptive fusion bands, it could be worth to check day/night samples separately.

Specific comments:

1. Line 10, Page 7: What the 13.3 channel is not used in the cloud mask detection?

2. Line 15 Page 7: Figures 7 and 8 in Wang et al. 2016 [doi.org/10.1002/2015JD024526] shows the importance of 13.3 and 6.7 channels for difference cases.

3. Line 3, Page 9: I think a 4 degree difference is too large for cloud comparisons. Do you mean 4 km?

4. Line 20, Page 9: How do you define pixel-level cloud fraction here, please clarify.

5. Line 2, Page 10: Could you please give the pixel fraction that CALIOP COTs are less than 0.03?

6. Line 3, Page 10: And it would be helpful if you can provide the cloudy and clear fractions in Table 2.

7. Line 19, Page 10: This is true. However, the authors could apply the same comparison to nighttime pixels to highlight the importance of water vapor and CO2 channels.

8. Table 2: What's the reason that the no fusion cloud mask retrievals are so different between NOAA-20 and SNPP in Arctic (e.g., 74.7% vs. 61.9%)?

9. Section 3.2, Page 12: How do you deal with multi-level clouds and mixed-phase

cloud? Did you use the uppermost cloud layer phases from CALIOP, in multiple cloud-layer cases? Please give more details.

10. Line 11, Page 16: In Figure 5, it is interesting that the fusion cloud-top heights (SNPP) are more negatively biased than no fusion heights in Antarctic. Do you have any speculation? I don't find the same feature in Figure 7 for NOAA-20.

11. Line 13, Page 22: Do you think it's due to artifacts of fusion bands? Since Figure 8c shows that near north pole, passive cloud-top height with fusion bands are higher than Lidar.

---

## Author Comment (AC1) · 14 May 2020

The present manuscript describes and validates the improvement of cloud retrievals from the VIIRS instrument on board of Suomi-NPP platform using radiances from CrIS hyperspectral instrument on board of the same platform.

The authors, using a fusion methodology, extracted broadband channel information from CrIS spectrally resolved measurements to simulate MODIS channels around 15 micron and 6.7 micron. In this way they can apply methodologies developed for MODIS to VIIRS that don't cover these spectral bands for cloud detection and retrieval. This improvement is been validated with CALIPSO dataset.

The manuscript topic is for sure appropriate for the Journal but in the present form has some incompleteness that should be fixed before publication. Incompleteness can be identified divided into two main topics: Hyperspectal instruments and Validation.

 • Regarding hyperspectral instruments as I said, in this work the authors use the spectrally resolved measurements of CrIS to simulate moderate resolution channels. In doing this the authors omitted to describe and acknowledge the great diagnostic power inside the spectral resolution and coverage of instruments like CrIS. For a reader who is not an expert in the field, it might appear that CrIS (and all the hyperspectral instruments) is a less accurate instrument than VIIRS because it has a worse spatial resolution. As an example, consider sentence at lines 15-21 of page 2 and lines 1-5 of page 3. It seems that CrIS has channels at 15 and 6.7 micron, missing in VIIRS instrument, but with degraded spatial resolution. I think that the authors should spend a sentence to indicate the peculiarities of hyperspectral instruments and add a figure showing a typical CrIS measurement in comparison with the spectral coverage of the channels used in the methodology described in the manuscript. Moreover I wish to recall that already 15 years ago it has been shown that with hyperspectral observation alone in the atmospheric window between 800-900 cm−1 is possible to detect and classify clouds. The authors can find an example in the following papers doi:10.1364/AO.41.000965 and doi:10.1016/S0022-4073(02)00083-3.

**Response: We made changes to better describe CrIS, which is a highly calibrated hyperspectral sounder. As the reviewer notes, the spatial resolution is much larger than that for VIIRS. Our methodology bridges this gap between the two sensors.**

For the figure showing a typical CrIS measurement in comparison with channels used in this study, please refer to Fig. 1 in Weisz et al. (2017):

Weisz, E., B. A. Baum, and W. P. Menzel, 2017: Construction of high spatial resolution narrowband infrared radiances from satellite-based imager and sounder data fusion. J. Appl. Remote Sens. 11 (3), 036022, doi: 10.1117/1.JRS.11.036022

We also included discussion of using the sounder for cloud detection and added the two papers in the Reference. New text reads "Previous studies detected the presence of clouds and retrieved cloud top height directly from sounder data (Masiello et al. 2002, 2003; Susskind et al. 2003; Li et al. 2005; Kahn et al. 2007)".

• About the validation, I have some doubts regarding the spatial distance between VIIRS and CALIPSO used for the colocation. While on the one hand I can imagine that a distance of less than 4◦ can reduce the concomitances between the two instruments, on the other a distance of 200 kilometers make the difference in spatial resolution between VIIRS and CrIS practically not appreciable. Probably a sentence that best justifies this choice is necessary. Also in relation to the results of the validation itself.

**Response: In addition to the spatial distance constraint, we also adopt other constraints to ensure collocations are appropriate between the two satellites. These include time differences, a sensor zenith threshold, a parallax correction, and minimum counts of collocations. This was also described in Heidinger et al. (2019) where the same collocation technique was used.**

**In replying to the reviewer's concerns, we ran collocations using a tighter spatial difference of 0.1 degree and presented the results for NOAA-20 below. It can be seen that while the counts decrease, the bias, standard deviation, and mode do not vary much compared to Table 5.**

**As suggested, we have added discussion as follows: "This approach allows maximum collocations between the two sensors, particularly in the polar regions. Though a large spatial distance is used, nearly all collocations (>99% globally) occur within 0.5° and about 60% of collocations are within 0.1°. We also note that use of tighter temporal and spatial thresholds does not impact the results significantly."**

**Similar as Table 5, but using collocation with a spatial difference within 0.1 degree.**

| Emissivity | | Counts | Bias (km) | Standard Deviation (km) | Mode (km) |
|---|---|---|---|---|---|
| 0 to 0.4 | No fusion | 19043 | -2.27 | 1.98 | -2.75 |
| | With fusion | | -1.99 | 1.77 | -2.25 |
| 0.4 to 0.8 | No fusion | 5578 | -1.91 | 1.51 | -1.75 |
| | With fusion | | -1.47 | 1.20 | -1.25 |
| 0.8 to 1.0 | No fusion | 73874 | -1.12 | 1.11 | -1.25 |
| | With fusion | | -1.04 | 1.07 | -1.25 |

• Page 2. Line 19. As I said before, CrIS has not only channels MODIS-like at 6.7 and 15 microns, but it covers the spectral ranges that MODIS cover with two channels with thousand channels.

**Response: We added the following text to the introduction: "In general, a sounding sensor is used for retrieving accurate atmospheric temperature and moisture profiles based on its hyperspectral coverage but at a lower spatial resolution than an imager such as VIIRS. CrIS takes measurements at 1305 wavelengths from 3.92-μm to 15.38-μm. The products from the CrIS sounder show significant enhancement over NOAA's legacy HIRS sensors."**

• Page 4. Line 11. Remove absorption before channel.
**Response: Done**

• Page 4. Line 19. The step (b) of the fusion method is not clear. The convolved sounder radiances are already at coarser spatial resolution. In the text it seems that the authors further degraded spatial resolution. Please clarify.

**Response: The convolved sounder radiances are derived for each CrIS field of view (FOV), i.e., at the CrIS native resolution. The basis of our technique is to derive a relationship between the imager 11/12-μm radiances and the average of the imager 11/12-μm pixel radiances within a given CrIS FOV. We do not degrade the CrIS spatial resolution further in this step, but simply average the VIIRS 11/12-μm radiances for all the pixels that lie within each of the CrIS FOVs. This is simply part of the k-d tree search methodology for determining how to best select the CrIS FOVs that should be used for each of the VIIRS**

**pixels to construct the IR absorption band radiances. For more details, please refer to Weisz et al. (2017).**

• Page 6. Line 16. Please insert a reference to the ACHA algorithm. If not, please place here the reference to the ATBD now at Page 7, line 2)
**Response: As suggested, we moved the reference to the ATBD to where ACHA first appeared.**

For these reasons I suggest to accept this manuscript subject to minor but necessary revisions.

---

## Author Comment (AC2) · 14 May 2020

General comments: Retrieval of cloud-top properties with the Visible Infrared Imaging Radiometer Suite (VIIRS) could be more challenging than its predecessor MODIS, because of the lack of water vapor and $CO_2$ bands in thermal infrared region. This paper "Improvement in cloud retrievals from VIIRS through the use of infrared absorption channels constructed from VIIRS-CrIS data fusion" by Li et al. demonstrated that by leveraging fusion water vapor and $CO_2$ bands from high-spectral resolution instrument CrIS, VIIRS cloud retrievals, including cloud mask, cloud thermodynamic phase, and cloud-top height are generally improved. This paper also shows that those fusion bands have a big boost in the accuracies of cloud mask/phase algorithm at high latitude. By including the extra fusion bands, cloud-top height retrieval is also improved with lower biases and uncertainties, in particular for those optically thin cirrus clouds with emissivity less than 0.8.

This paper is well organized and written. One of my major concerns is that the authors should give more details about the comparisons between VIIRS retrievals and CALIPSO/CALIOP. Furthermore, to highlight the importance of those absorptive fusion bands, it could be worth to check day/night samples separately.
**Response: We appreciate your comments and address specific comments as noted below.**

**Please note that we studied day/night samples separately for cloud mask, phase and height. Results for cloud mask are presented in response to Comment 7. For cloud phase, the general conclusion is similar and the primary difference between day and night is detecting more water phase clouds during day because of an additional test by the VIIRS 1.6um channel, which also results in slightly larger increase in the percentage of correctly identified ice phase clouds compared to nighttime when fusion channels are used. For cloud height products, since only IR channels are used in ACHA and cloud phase is matched in the validation, no obvious differences are observed. Relevant discussions have been added to the manuscript.**

Specific comments:
1. Line 10, Page 7: What the 13.3 channel is not used in the cloud mask detection?
**Response: The cloud mask team led by one of the coauthors here, Dr. Andrew Heidinger, conducted cloud detection tests using various spectral channels. It was found that adding**

**the 13.3um channel did not help as much as the 6.7um channel. For the way our cloud mask is constructed, one doesn't need both channels.**

2. Line 15 Page 7: Figures 7 and 8 in Wang et al. 2016 [doi.org/10.1002/2015JD024526] shows the importance of 13.3 and 6.7 channels for difference cases.

**Response: We have revised the sentence "It is difficult to explain definitively the information content available in each of these IR bands so the approach is to test their impact on ice cloud height retrievals..." Now it reads "Previous studies explored spectral band information useful for cloud property retrievals by computing the Shannon information content (L'Ecuyer et al. 2006, Wang et al. 2016). The approach used here is to test their impact on ice cloud height retrievals through comparison with another cloud height product."**

3. Line 3, Page 9: I think a 4 degree difference is too large for cloud comparisons. Do you mean 4 km?

**Response: This has been addressed in response to Referee #1's comment.**

4. Line 20, Page 9: How do you define pixel-level cloud fraction here, please clarify.

**Response: We added discussion to clarify how we define pixel level cloud fraction: "When a cloud layer is detected by CALIPSO/CALIOP, the pixel is classified as cloudy. Neighboring pixels along the path are included and the cloud fraction is defined by computing the ratio between the number of cloudy pixels and the total number of pixels."**

5. Line 2, Page 10: Could you please give the pixel fraction that CALIOP COTs are less than 0.03?

**Response: A table is shown below of counts and fraction of CALIOP COTs less than 0.03. We added a sentence as follows: "The fraction of the sub-visible clouds is less than 4% from a global perspective and less than 3% in the polar regions."**

| | | Sample Size COT < 0.3 | Sample Size All | Ratio |
|---|---|---|---|---|
| **S-NPP** | **Global** | **217983** | **6091230** | **0.036** |
| | **60°N to 60°S** | **176734** | **4384193** | **0.040** |
| | **Arctic** | **16968** | **853006** | **0.020** |
| | **Antarctic** | **24281** | **854031** | **0.028** |
| **NOAA-20** | **Global** | **73869** | **2328596** | **0.032** |
| | **60°N to 60°S** | **58975** | **1645684** | **0.036** |
| | **Arctic** | **10374** | **329702** | **0.031** |
| | **Antarctic** | **4520** | **353210** | **0.013** |

6. Line 3, Page 10: And it would be helpful if you can provide the cloudy and clear fractions in Table 2.

**Response: We added the numbers of cloud fractions from both sensors to Table 2 and the following discussion: "In terms of total cloud fraction, as expected, VIIRS tends to report a lower cloud fraction than CALIOP. CALIOP has a better detection sensitivity to optically thin clouds, and global cloud fractions reported from the two sensors are in agreement when the minimum cloud optical thickness is set between 0.6 and 0.7. The global values do not necessarily become more closely aligned with CALIOP when a fusion channel is used. However, the use of a fusion channel results in a much larger impact in the polar regions, as will be shown in Figure 1."**

7. Line 19, Page 10: This is true. However, the authors could apply the same comparison to nighttime pixels to highlight the importance of water vapor and CO2 channels.

**Response: We are unsure if the reviewer is referring to this sentence "This is unsurprising since the cloud mask algorithm performs fairly well for a snow-free surface...". As noted, the cloud mask algorithm is not using the CO2 channels. We are computing the validation of cloud mask detection separating day and night as requested below. Note that we used a solar zenith angle threshold of 85 degrees to separate day and night, and discarded pixels that do not have a valid solar zenith angle.**

**Daytime Only**

| | | **Sample Size** | | **Correct Detection** | **Missed Cloud** | **False Detection** |
|---|---|---|---|---|---|---|
| **S-NPP** | **Global** | **2899130** | **Fusion** | **82.9** | **13.1** | **3.9** |
| | | | **No Fusion** | **82.4** | **13.6** | **3.9** |
| | **60°N to 60°S** | **2154403** | **Fusion** | **84.3** | **12.5** | **3.1** |
| | | | **No Fusion** | **84.1** | **12.8** | **3.1** |
| | **Arctic** | **469177** | **Fusion** | **77.4** | **15.0** | **7.6** |
| | | | **No Fusion** | **75.5** | **16.7** | **7.8** |
| | **Antarctic** | **275550** | **Fusion** | **80.9** | **14.7** | **4.4** |
| | | | **No Fusion** | **80.7** | **15.2** | **4.1** |

| | | | | Correct Detection | Missed Cloud | False Detection |
|---|---|---|---|---|---|---|
| **NOAA-20** | **Global** | **1098695** | **Fusion** | **84.4** | **12.5** | **3.1** |
| | | | **No Fusion** | **83.9** | **13.1** | **3.0** |
| | **60°N to 60°S** | **799995** | **Fusion** | **84.8** | **12.1** | **3.1** |
| | | | **No Fusion** | **84.6** | **12.4** | **3.1** |
| | **Arctic** | **14939** | **Fusion** | **84.9** | **10.0** | **5.1** |
| | | | **No Fusion** | **80.9** | **11.4** | **7.7** |
| | **Antarctic** | **283761** | **Fusion** | **83.3** | **13.8** | **2.8** |
| | | | **No Fusion** | **82.2** | **15.4** | **2.4** |

**Nighttime Only**

| | | **Sample Size** | | **Correct Detection** | **Missed Cloud** | **False Detection** |
|---|---|---|---|---|---|---|
| **S-NPP** | **Global** | **2974117** | **Fusion** | **83.6** | **11.9** | **4.5** |
| | | | **No Fusion** | **82.6** | **12.0** | **5.4** |
| | **60°N to 60°S** | **2053056** | **Fusion** | **87.2** | **8.8** | **4.0** |
| | | | **No Fusion** | **87.1** | **8.7** | **4.2** |
| | **Arctic** | **366861** | **Fusion** | **76.3** | **16.0** | **7.7** |
| | | | **No Fusion** | **73.7** | **17.0** | **9.3** |
| | **Antarctic** | **554200** | **Fusion** | **75.3** | **20.5** | **4.2** |
| | | | **No Fusion** | **71.8** | **21.0** | **7.2** |

| | | | | | | |
|---|---|---|---|---|---|---|
| **NOAA-20** | **Global** | **1156032** | **Fusion** | **81.2** | **13.7** | **5.1** |
| | | | **No Fusion** | **79.6** | **13.4** | **7.0** |
| | **60°N to 60°S** | **786714** | **Fusion** | **86.2** | **9.4** | **4.4** |
| | | | **No Fusion** | **86.1** | **9.2** | **4.7** |
| | **Arctic** | **304389** | **Fusion** | **66.7** | **25.7** | **7.5** |
| | | | **No Fusion** | **60.9** | **25.0** | **14.0** |
| | **Antarctic** | **64929** | **Fusion** | **89.1** | **8.8** | **2.1** |
| | | | **No Fusion** | **88.2** | **10.0** | **1.8** |

8. Table 2: What's the reason that the no fusion cloud mask retrievals are so different between NOAA-20 and SNPP in Arctic (e.g., 74.7% vs. 61.9%)?

**Response: The data used for NOAA-20 and SNPP in this study are from different seasons, so this could be playing a part. The SNPP data are from April and October 2018, while NOAA-20 data are from January 2019. Given the limited amount of data processed for this study, we need to further investigate this difference more closely.**

9. Section 3.2, Page 12: How do you deal with multi-level clouds and mixed-phase cloud? Did you use the uppermost cloud layer phases from CALIOP, in multiple cloudlayer cases? Please give more details.

**Response: CLAVR-x does not retrieve mixed-phase cloud. There is some logic for discriminating the presence of multilayered clouds (primarily optically thin ice clouds overlying a lower-level liquid water cloud), and these are treated as ice phase in this study. We note that the uppermost cloud layer phase from CALIOP in multilayer cases is used.**

10. Line 11, Page 16: In Figure 5, it is interesting that the fusion cloud-top heights (SNPP) are more negatively biased than no fusion heights in Antarctic. Do you have any speculation? I don't find the same feature in Figure 7 for NOAA-20.

**Response: The bias is small in the Antarctic, and we are not sure what caused this behavior. We need to process much more data over seasons to determine whether this is caused by a relatively high surface elevation or some other factor.**

11. Line 13, Page 22: Do you think it's due to artifacts of fusion bands? Since Figure 8c shows that near north pole, passive cloud-top height with fusion bands are higher than Lidar.

**Response: There is an indication that the cloud heights improve further with the addition of the 6.7um fusion channel. What is interesting about this is that the 6.7-μm channel is not generally used for global operational cloud height retrievals; the 13.3-μm channel is more often used. The 6.7-μm channel is strongly impacted by the presence of water vapor, and obviously the amount of water vapor is quite small in the Antarctic. We need to do further study to determine the information content of this channel at high latitudes.**